

# On the 2011 record low Arctic sea ice thickness: a combination of dynamic and thermodynamic anomalies

Xuewei Li[1], Qinghua Yang[1], Lejiang Yu[2], Paul R. Holland[3], Chao Min[1], Longjiang Mu[4], Dake Chen[1]

[1]School of Atmospheric Sciences, Sun Yat-sen University, and Southern Marine Science and Engineering Guangdong
Laboratory (Zhuhai), Zhuhai 519082, China
[2]MNR Key Laboratory for Polar Science, Polar Research Institute of China, Shanghai 200136, China
[3]British Antarctic Survey, Cambridge CB3 0ET, United Kingdom
[4]Qingdao National Laboratory for Marine Science and Technology, Qingdao 266237, China

*Correspondence to*: Qinghua Yang (yangqh25@mail.sysu.edu.cn)

**Abstract.** The sea ice thickness is recognized as an early indicator of climate changes. The mean Arctic sea ice thickness has been declining for the past four decades, and a sea ice thickness record minimum is confirmed occurring in autumn 2011. We used a daily sea ice thickness reanalysis data covering the melting season to investigate the dynamic and thermodynamic processes leading to the minimum thickness. Ice thickness budget analysis demonstrates that the ice thickness loss is associated with an extraordinarily large amount of multiyear ice volume export through the Fram Strait during the season of sea ice advance. Due to the loss of multiyear ice, the Arctic ice thickness becomes more sensitive to atmospheric anomalies. The positive net surface energy flux anomalies melt roughly 0.22 m of ice more than usual from June to August. An analysis of clouds and radiative fluxes from ERA5 reanalysis data reveals that the increased net surface energy absorption supports the enhanced sea ice melt. The enhanced cloudiness led to positive anomalies of net long-wave radiation. Furthermore, the enhanced sea ice melt reduces the surface albedo, triggering an ice–albedo amplifying feedback and contributing to the accelerating loss of multiyear ice. The results demonstrate that the dynamic transport of multiyear ice and the subsequent surface energy budget response is a critical mechanism actively contributing to the evolution of Arctic sea ice thickness.

## 1 Introduction

Arctic sea ice plays an important role in the climate system. The thinning of Arctic sea ice well reflects recent climate changes. The submarine and satellite (ICESat) records reveals a long-term trend of Arctic sea ice thinning since 1958 to 2008(Kwok and Rothrock, 2009; Rothrock et al., 2008). Along with the observed decrease in sea ice thickness, multiyear ice (MYI) has been shrinking faster than the entire sea ice (Comiso, 2002; Kwok and Untersteiner, 2011). The fraction of multiyear sea ice in the total ice extent in March decreased from about 75% in the mid 1980s to 45% in 2011 (Maslanik et al., 2011). Most notably, younger and thinner ice becomes more sensitive to dynamic and thermodynamic effects such as ice drift and melting (Maslanik et al., 2007; McPhee et al., 1998). The Arctic sea ice thickness declined sharply in 2005 and 2007, while the thinning trend deaccelerated in the years of the CryoSat-2 record (Kwok, 2018). Min et al. (2019) claimed





that the minimum sea ice thickness occurs in 2011, using a model combined with satellite thickness data, and this is also found in satellite data alone (Kwok and Cunningham, 2015; Tilling et al., 2015). The seasonal evolution of mean Arctic sea ice thickness (SIT), volume (SIV) and area from October 20, 2010 through February 22, 2020 are contrasted in Figure 1. The

datasets based on European Space Agency (ESA) and Alfred Wegener Institute Helmholtz Centre for Polar and Marine Research (AWI) CryoSat-2 satellite data are tracked between mid-October and mid-May (Kurtz and Harbeck, 2017; Ricker et al., 2014). The mean sea ice thickness within the area of actual ice coverage in October 2011 reached the lowest record for that calendar month in any year of the satellite records, even though the SIV minimum in October 2011 proved to be only the third lowest on record, eclipsed by values in 2019 and 2020 (Fig. 1).

Different theories have been proposed to explain the mechanism of Arctic sea ice thickness loss. The sea ice thickness distribution is determined by the thermodynamic and dynamic response of sea ice to atmospheric and oceanic forcings (Thorndike et al., 1975), so sea ice thickness loss is associated with the changes in atmosphere, ocean, and sea ice circulation (Comiso et al., 2008; Lindsay and Zhang, 2005; Smedsrud et al., 2017). Sea ice motion is mainly wind driven (Lindsay et al., 2009). Laxon et al. (2003) revealed that the high-frequency interannual variability in mean Arctic ice

thickness is dominated by changes in the amount of summer melt. As the Arctic atmosphere warms, the summer melting season extends, the oceans absorb more heat and thus the winter freeze is delayed (Stroeve et al., 2012). Ricker et al. (2017) revealed that the loss of ice volume was associated with a reduction in the cumulative freezing degree days. Some evidences suggest that there are significant anomalies in the Arctic in 2011. The analysis of Ogi and Wallace (2012) showed that low level winds over the Arctic in 2011 play an important role in mediating the rate of decrease of sea ice extent during summer.

From mid-June in 2011, the melt pond fraction exhibits values up to two standard deviations above the mean values for the years 2000–2011, which are even higher than in summer 2007 (Rösel and Kaleschke, 2012). To advance upon these existing studies, we utilized sea ice thickness budgets to further assess the dynamic and thermodynamic mechanisms involved in the sea ice thickness anomaly in summer 2011.

In this study, we address two major questions by sea ice budget analysis. First, what special factors led to the precipitous

decrease of sea ice thickness in 2011 and produced the record low? Second, are the changes predominantly dynamic or thermodynamic in origin? The paper is organized as follows: Sect. 2 describes the observations and model data used in this study and presents the methods that we employed to investigate the sea ice budgets. In Sect. 3, we evaluate the sea ice budget anomalies in 2011. Moreover, the mechanism for sea ice thinning in response to the driving climatic factors are described. We summarize and discuss the major findings of this study in Sect. 4.

**2 Data and methods**

The evolution of sea ice thickness is governed by the dynamic and thermodynamic processes. The thickness can be separated by a simple conservation equation ( e.g. Bitz et al., 2005; Holland et al., 2014)

$$\frac{\partial H}{\partial t} = -\nabla \cdot (uH) + residual \,, \tag{1}$$





$$= -u \cdot \nabla H - H\nabla \cdot u + residual \ , \tag{2}$$

where $H$  is sea ice thickness and $u$ is ice motion. The term on the left-hand side $\frac{\partial H}{\partial t}$ is referred to as ice "thickening", which

is determined by ice thickness flux divergence, $\nabla \cdot (uH)$ and the residual. The flux divergence can be separated into

"advection", $u \cdot \nabla H$, and "divergence", $H\nabla \cdot u$. The residual represents the thermodynamic melting and freezing. We adopt

the sign convention that positive values of all terms are associated with an increase in ice thickness. In a related approach,

Holland and Kimura (2016) examine the Arctic ice concentration budget terms, which is highly instructive, but our purpose

is to assess the ice thickness budget. The sea ice thickness is defined as grid cell-averaged ice thickness, which is also called

effective ice thickness. The effective ice thickness is the product of the average ice thickness and the ice area concentration

and equals the volume of ice per unit area of ocean.

We apply this methodology to a well-validated sea ice thickness and drift dataset (the Combined Model and Satellite

Thickness data, CMST), which was generated by the MITgcm ice-ocean model with CryoSat2, SMOS sea ice thickness and

SSMIS sea ice concentration assimilated (Mu et al., 2018). The CMST thickness data cover both the cold seasons and the

melting seasons for the period of October 2010 to December 2016 on an 18-km grid. The CMST has been already

quantitatively evaluated against observations by a previous study (Min et al., 2019; Mu et al., 2018), demonstrating an

accurate performance in simulating the real sea ice drift and thickness. To reduce the noise in ice drift fields and hence

divergence calculation, we follow Holland and Kimura (2016) and smooth ice drifts with a 400×400 km square-window

filter.

To evaluate sea ice variability, we use the Arctic sea ice thickness and concentration data based on ESA and AWI CryoSat-2

satellite (Kurtz and Harbeck, 2017; Ricker et al. 2014). The ESA ice thickness data are provided daily from October 2010 to

April 2020, while the AWI ice thickness are provided weekly. We also use the weekly sea ice age for the Arctic Ocean

(Tschudi et al., 2020). The method used to estimate sea ice age involves Lagrangian tracking of sea ice from week-to-week

using gridded ice motion vectors (Maslanik et al., 2011; Tschudi et al., 2019).

We also use the sea ice thickness and drift in CMST data to compute ice volume export through Fram Strait. We follow the

previous definition of gate position and defined the gate at 82° N between 12° W and 20° E and 20° E between 80.5 and 82° N

(Krumpen et al., 2016). Because the EASE-Grid Sea Ice Age product is not provided near coasts, the sum of FYI and MYI is

slightly less than the total amount of ice.

In this study, to quantify the thermodynamic impact on the ice thickness budget, we estimate sea level pressure (SLP), 10 m

wind speed, surface radiation fluxes, and albedo anomalies, derived from monthly ERA5 atmospheric reanalysis data from

the European Centre for Medium-Range Weather Forecasts (ECMWF)(Copernicus Climate Change Service (C3S), 2017;

Hersbach et al., 2020).





## 3 Results

### 3.1 Sea ice thickness budget anomalies

In this section, we first analyze the seasonal sea ice budget in Eq. 2 to evaluate the thermodynamic and dynamic impacts on sea ice evolution (Figure 2). According to the timing of ice advance versus ice retreat, we time-integrated the Arctic sea ice thickness budget into seasonal means for the seasons of sea ice advance (October to April) and retreat (May to September). The Arctic sea ice thickening increased (declined) rapidly from October to April (May to September) (Figure 2a,e). From

October to April, advection extended the ice equatorward at the ice edge, most notably in the Beaufort Sea (BS) and east of Greenland (Figure 2b). Persistent strong divergence occurred in Central Arctic (CA), with sea ice convergence along the coasts of the BS, Chukchi Sea (CS) and East Greenland (Figure 2c). These dynamical ice transports lead to strong thermodynamic growth throughout most of the Arctic, but melting around Greenland (Figure 2d). As demonstrated by Bitz et al. (2005) and Holland and Kimura (2016), this melting occurs because ice is advected into regions where warmer oceanic

and atmospheric conditions make the ice thermodynamically unsustainable, even in winter.

From May to September, thermodynamic ice loss dominates the budget, while the pattern of dynamic advection and divergence remains consistent with that in October-April. Overall, during the seasons for both advance and retreat, the residual was dominated by thermodynamics, with widespread melting from May to September and freezing in other months (Figure 2d,h). The dynamics play an important role, however, which is strongest from October to April.

From October 2010 to September 2011, the entire Arctic lost 1278 $km^3$ of sea ice, with the central Arctic accounting for 52% of the loss (Fig. 1). Notably, the Arctic lost 1078 $km^3$ of multiyear ice in 2011, accounting for 84% of the total sea ice loss. To determine the origin of anomalies in sea ice thickness, we analyze the Arctic sea ice thickness budget anomalies from October 2010 to September 2011 by subtracting the 6-year mean from each month (Figure 3). It is important to note that each term in Figure 3 represents the contribution of dynamic or thermodynamics processes to sea ice thickness anomalies.

Generally, the total sea ice thickening anomaly was negative in 2011 which indicated that more sea ice was being lost in the whole Arctic region, especially in the north of Canadian Arctic Archipelago (CAA) (Fig. 3).

During the season of sea ice advance, the thickening anomaly is relatively weak. The negative thickening anomaly (approximately -0.1 meters per month) appeared along the north of CAA and the coast of the East Siberian Sea (ESS). Most regions were subject of enhanced divergence, while increased convergence is indicated around the coast of the BS.

Advection anomalies transported sea ice from BS, CS, and ESS to the Fram Strait along the CA coast, resulting in increased sea ice thickness north of Spitsbergen and the Fram Strait. Residual (thermodynamic) anomalies were relatively weak but matched the overall thickening anomalies through most of the Arctic, resulting in enhanced ice thickening north of the BS and CS and reduced ice thickening in ice thickness in the north of CAA. The dynamic anomalies around north and east Greenland induced residual thermodynamic changes. Increased ice advection east of Greenland caused enhanced ice melting

and hence an anomalous freshwater flux to the ocean, while increased divergence north of Svalbard induced greater freezing.





These changes demonstrate a stabilising thermodynamic feedback that responds to the dynamic anomalies, as neither change is reflected in the overall ice thickening anomalies.

During the season of sea ice retreat, strong negative thickening anomalies were dominated by the residual processes, which means that the thermodynamics played a greater role in the summer ice retreat. In the regions where multi-year ice exists
along the BS and CA coasts, a strong sea ice thinning is found, only very weakly offset by convergence and advection (Figure 3f,g). There was a strong correspondence between thickening and residual anomalies in CA, which we interpret as changes in melting directly induced by anomalies in atmospheric temperature, net surface heat flux, and other variables. As shown in Figure 4d, the mean surface net heat flux of entire Arctic from June to August is more than the mean values from 2011 to 2016 by up to 4 W/m². At the same time, the residual anomalous heat flux increased from June to August, leading to
a significant sea ice loss.

### 3.2 Dynamic transportation of the Sea Ice Anomaly

We have obtained monthly AO indices from NOAA (http://www.cpc.ncep.noaa.gov). In 2010, there was a strong and sustained negative AO phase that had not been seen since the late 1960s (Stroeve et al., 2011). However, from February to April 2011, SLP patterns reverted to the high positive phase of AO (Fig.4a). Thus, in order to investigate the thickness
anomalies in terms of ice dynamics, we assessed seasonal mean NSIDC ice drift anomaly according to the AO phase (Fig.5). From October to January, peak SLP anomalies (of 12 hPa) were centered south of Greenland (Fig.5g). At the same time, there was a weak negative SLP anomaly in eastern Siberia. The SLP anomalies led to a gradual divergence of sea ice from the east Arctic toward the north of the CAA (Fig.5d). The enhanced motion along the Transpolar Drift Stream was broadly parallel to the anomalous SLP gradient. The enhanced transpolar advection transported sea ice from the BS and the coast of
CA to the Fram Strait (Fig.5a). Compared with the average from 2010 to 2016, the amount of multiyear ice exported through the Fram Strait increased by 219 km³ from October to January (Fig.4b).

From 2011 February to April, an AO index of more than 1.7 produced negative SLP anomalies of 10–15 hPa (Fig.5h). The positive AO phase weakened the Beaufort Sea High, promoting a cyclonic atmospheric circulation anomaly, which weakened the ice motion from the east Arctic towards the CA, resulting in the enhancement of ice thickening in the BS and
CS (Fig.5b,e). Under these cyclonic surface wind stress anomalies, Ekman transport deflected ice drift to the right, into the coast. This process decreases the recirculation of ice and increases the ice divergence over the Arctic (Figure 5e). The surface wind anomalies decreased the Transpolar Drift Stream, leading to reduced ice transport out of the Arctic Ocean through Fram Strait in February and April. However, in March, negative SLP anomalies shifted toward the eastern Arctic Ocean, and the enhanced Transpolar Drift Stream exported an additional 116 km³ of multiyear ice through Fram Strait
(Fig.4b), contributing to the thinning of sea ice along the north of the CAA. In March, the surface air temperature exhibited pronounced warming (Fig.4c) over the eastern Arctic, but less in the western Arctic. The correlation between wintertime surface air temperature and the AO seems broadly consistent with the mechanism outlined by Rigor et al.(2002). The AO





drives sea ice to dynamically thin, resulting in an enhanced heat flux from the ocean, and raising the surface air temperature. The loss of multiyear ice, together with warmer surface air temperature, leads to further loss of sea ice.

From May to August, the AO index turned from positive to negative phase. While the mean AO index in May to August was −0.6, peak SLP anomalies (of 8 hPa) were centered over the CA, implying enhanced Ekman convergence (Figure 5f,i). The Beaufort Gyre was stronger than climatological values, leading to enhanced ice transport from the south to the north Arctic and from the East Siberian Sea to Fram Strait.

It can be seen from Figure 5 that dynamic ice thickening anomalies due to ice drift anomalies presented an opposite pattern

between February-April and May-August. This was mainly manifested as opposing anomalies in many regions, including the CA, BS and eastern Arctic (CS, ESS, LS). In February-April, the divergence in CA enhanced, resulting in the decrease of ice thickness in CA and the enhanced ice thickness in BS, CS and eastern Arctic (LS, KS). The dynamical influence in May-August was the opposite. The main reason for this change was the transformation of AO index from positive to negative mode.

The total sea ice and multiyear sea ice fluxes through the transect at 82ºN from October to April are 1546±160 km$^3$ and 1082 km$^3$, respectively (Table 1). Compared with the average from 2010 to 2016, an additional 233 km$^3$ of multiyear sea ice were exported through Fram Strait during the season of ice advance in 2011. Although the data used are different, previous studies have also shown an abnormal increase in sea ice fluxes through the Fram Strait in 2011 (Min et al., 2019; Ricker et al., 2018). Hence, multiyear sea ice loss for the season of sea ice advance has contributed to the negative summer sea ice

thickness anomalies.

### 3.3 Thermodynamic forcing and thermodynamic feedback

In May-September, a negative thickening anomaly (Figure 3e) showed that more sea ice was lost in 2011. This was manifested through enhanced thermodynamic melt, hence a strong negative residual anomaly in CA (Figure 3h). The spatial pattern of residual anomaly was very similar to that of net surface energy flux (Figure 6,a-f). Persistent negative (i.e.

downward) net surface energy flux anomalies were found over most parts of the negative residual anomalies in June-August, which was evident from Fig.4d. The net surface energy flux provided 8.67 W·m$^{-2}$ more than normal values into sea ice during summer 2011 in CA.

The net surface heat budget of the Arctic Ocean is dominated by radiative fluxes, and the exchange of heat between the Arctic Ocean and atmosphere is strongly moderated by the thickness of sea ice (Bobylev and Miles, 2020; Untersteiner,

185  1964).

$$F_{net} = LW_{net} + SW_{net} - H_s - H_l \,, \tag{3}$$
$$= LW_d - LW_u + SW_d - SW_u - H_s - H_l, \tag{4}$$

$F_{net}$ is the net surface energy flux, $LW$ ($SW$) represent the longwave (shortwave) downwelling (d) and upwelling (u) fluxes, as well as sensible, $H_s$, and latent, $H_l$, heat fluxes.  When the ice becomes thick enough that no heat from the ocean can be





conducted through the ice, sea ice serves as an insulator, limiting sensible and latent heat transfers from the ocean (Maykut and Untersteiner, 1971; Untersteiner, 1964).

The negative net surface longwave radiation anomalies in June were found over most of the Arctic Ocean, except the BS and CS (Fig. 6g). The anomalies of the net longwave radiation were roughly the same as those of the downward component (not shown), contributed to the energy surplus at the surface. The cloud fraction had a moderately negative anomaly in BS and

CS (over 10%) and a positive anomaly (10%) in eastern Arctic in June (Fig. 6s). Corresponding to these anomalies in cloud amounts, these areas had strong anomalies of the longwave radiative forcing. In July, the reduced cloud cover over the Arctic (negative anomaly in Figure 6t) had led to a decrease in downward longwave radiation (positive anomaly in Figure 6h). The cloud cover increased again in August, so that the downward longwave radiation was enhanced which warmed the surface.

In addition to the absorption of the longwave, clouds can also reflect the sunlight. The net shortwave radiation showed a

negative anomaly at the west of Arctic in June (Fig. 6j). The negative cloud fraction anomaly at the west of Arctic resulted in up to a 40 W/m$^2$ anomaly in downward shortwave radiation. Although the pattern of downward short-wave radiation (not shown) was similar to that of cloudiness, the upward short-wave radiation anomaly in the BS, CS and ESS was offset by enhanced sea ice albedo in June. In July, the negative cloud area fraction anomalies reduced the solar energy reflected to space (Fig. 6t). At the same time, the change in albedo increased the short-wave radiation absorbed, which further promoted

melting and warming (Fig. 6k,w). In August, due to the increased cloud area fraction, the net short-wave radiation exhibited negligible positive anomalies. However, the melting of sea ice resulted in a decrease in albedo over most of the Arctic in August and contributed to an increase in net short-wave radiation absorption in CA.

The energy flux anomalies caused by net long-wave and short-wave radiation were only compensated by corresponding increases of the sensible heat fluxes to a minor extent, because the surface temperature cannot increase significantly above 0 ℃

with ice present; the bulk of the energy went into ice melting rather than warming of the surface. However, close to the ESS and LS the positive sensible heat flux anomalies increased (Fig.6o) indicating that the sea surface temperature is warmer than usual.

The net surface energy fluxes anomalies provided extra energy, beginning in June. In months of abnormal high cloud cover (June and August), the long-wave radiation generally dominated the surface radiation budget due to lack of sunlight. The net

surface energy fluxes were accomplished by the increased downward long-wave radiation and decreased downward short-wave radiation. In July, due to the reduced cloud cover, the downward solar radiation increased, while albedo feedback further amplified the net short-wave radiation anomaly. Hence the anomalies of cloud fraction played a significant role, whereas the albedo anomalies acted through an amplifying feedback process when the melting was already initiated.

## 4 Summary and Discussion

In this study, we confirm that the mean sea ice thickness in autumn 2011 hit the minimum of the satellite record, using sea ice thickness observations from ESA and AWI CryoSat-2 satellite and the CMST reanalysis data. Together with the satellite-



based sea ice motion data, we quantify the sea ice thickness budget in the Arctic between 2011 and 2016. These thickness anomalies result from the interplay between ice dynamic and thermodynamic processes.

Our findings suggest the driver for net ice mass loss in September 2011 was an anomalously low volume of MYI during the season of sea ice advance. The observed declining trends in MYI volume were associated with dynamic processes in October, January and March. Accelerated ice motion led to enhanced ice transport, which was associated with the Arctic Oscillation. An additional 233 km$^3$ of MYI was exported through Fram Strait during the season of ice advance. First-year ice, which was more sensitive to anomalies in the thermodynamic forcing, then replaced the multiyear ice. Our results further demonstrate that the thermodynamic forcing was determined by the persistent net surface energy flux anomaly over the Arctic, which played an important role in mediating the retreat of sea ice during summer. In June and August, positive cloud cover anomalies increased downward longwave radiation and reduced downward shortwave radiation over the Arctic. The opposite case occurred in July. In addition, the ice-albedo feedback also can modulate the net surface short radiation.

Our results highlight the importance of atmospheric forcing on sea ice thickness variability, as noted in previous studies (Maslanik et al., 2011; Ogi and Wallace, 2012; Rösel and Kaleschke, 2012). In order to improve our capability to predict the sea ice and understand the sea ice thickness variability and long-term trends, our understanding of the linking mechanisms between atmospheric forcing and sea ice thickness should be strengthened, and the atmosphere-sea ice-ocean observations collected from the very recent MOSAiC campaign (https://mosaic-expedition.org/expedition) will provide us a unique opportunity in this regard.

*Data availability.* The CMST sea ice data are publicly available at https://doi.pangaea.de/10.1594/PANGAEA.891475 (Mu et al., 2018). The ERA5 climate reanalysis data from the Copernicus Marine Environment Monitoring Service under https://cds.climate.copernicus.eu. The datasets based on European Space Agency (ESA) and Alfred Wegener Institute Helmholtz Centre for Polar and Marine Research (AWI) CryoSat-2 satellite data are available at https://nsidc.org/data/RDEFT4/versions/1 and http://www.meereisportal.de. The EASE-Grid Sea Ice Age data are provided by the National Snow and Ice Data Center (NSIDC, https://nsidc.org/data/NSIDC-0611/versions/4,). The AO index are provided by the National Centers for Environmental Prediction (NCEP) Climate Prediction Center at https://www.cpc.ncep.noaa.gov/products/precip/CWlink/ daily_ao_index/ao.shtml.

*Author contributions.* QY and XL designed the study, XL carried the analysis and wrote this manuscript. All authors participated in paper preparation, reviewed and edited this manuscript.

*Competing interests.* The Authors declare that they have no conflict of interests.

*Acknowledgments.* This work was supported by the National Natural Science Foundation of China (No. 41922044, 41941009,



41676185), the Guangdong Basic and Applied Basic Research Foundation (No. 2020B1515020025), the State Key Laboratory of Cryospheric Science (No. SKLCS-OP-2020-3,SKLCS-OP-2020-5), the Fundamental Research Funds for the Central Universities (No. 19lgzd07, 20lgpy24) and the China Postdoctoral Science Foundation (No. 2020M683022). This is a contribution to the Year of Polar Prediction (YOPP), a flagship activity of the Polar Prediction Project (PPP), initiated by
the World Weather Research Programme (WWRP) of the World Meteorological Organisation (WMO). We acknowledge the WMO WWRP for its role in coordinating this international research activity.

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





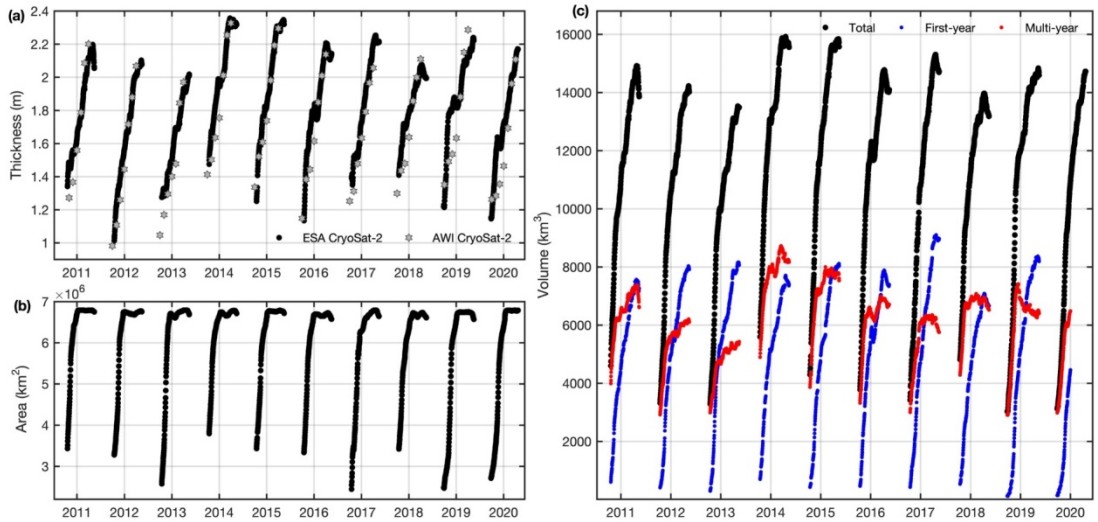

**Figure 1**: **Daily behaviour of sea ice volume, area and thickness based on ESA and AWI CryoSat-2 satellite datasets from October 2010 through April 2020. (a) Mean sea ice thickness within area of actual ice coverage. (b) Total sea ice area (cumulative area of actual ice coverage) within Arctic basin. (c) Total(black),first-year(blue) and multiyear (red) sea ice volumes within Arctic basin. Arctic basin volume and area is computed within the bounded by the gateways into the Pacific (Bering Strait), the Canadian Arctic Archipelago, and the Greenland (Fram Strait) and Barents Seas.**


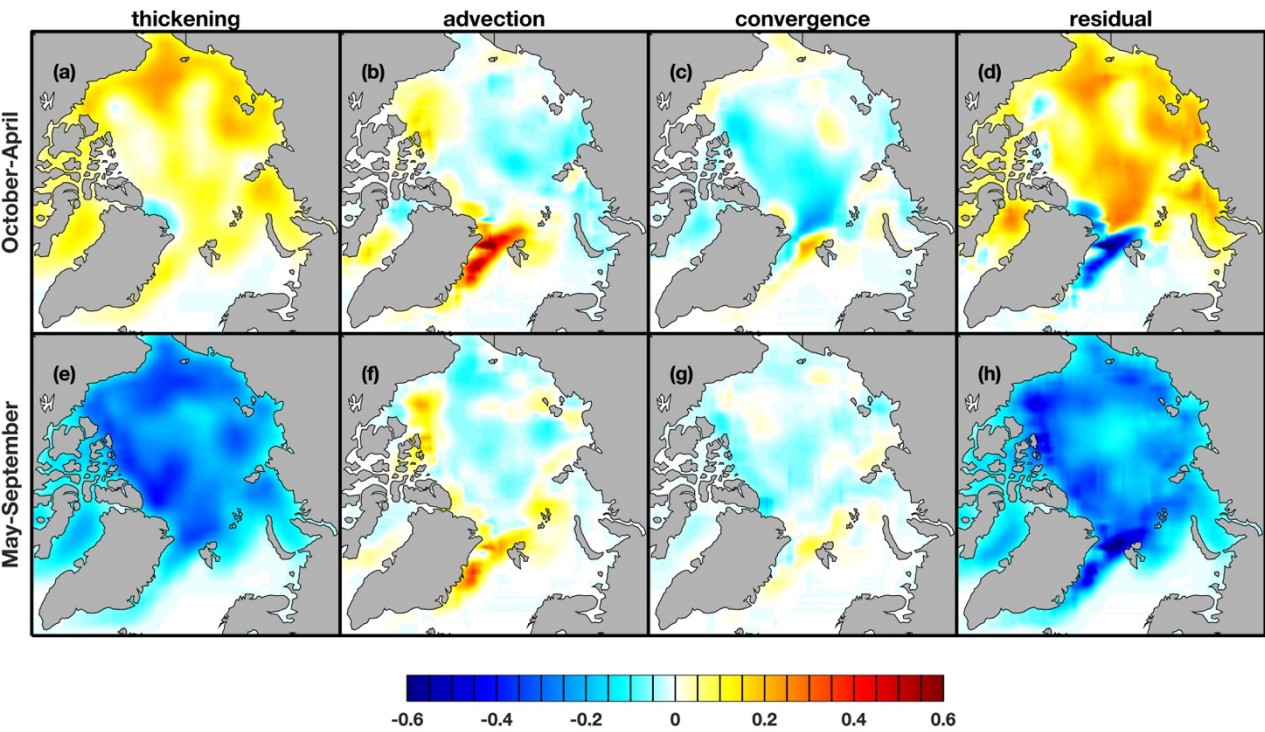


**Figure 2**: **Components of seasonal Arctic sea ice thickness budget from October 2010 to September 2011 (unit: meters per month). The top row shows the sea ice thickness budget during sea ice advance (from October to April) and the bottom row shows that during sea ice retreat (from May to September). (a,e) thickening, $\frac{\partial \Delta H}{\partial t}$; (b,f) advection, $-u \cdot \nabla H$ ; (e,g) convergence $-H\nabla \cdot u$;**

**(d,h) residual, $f_H$.**






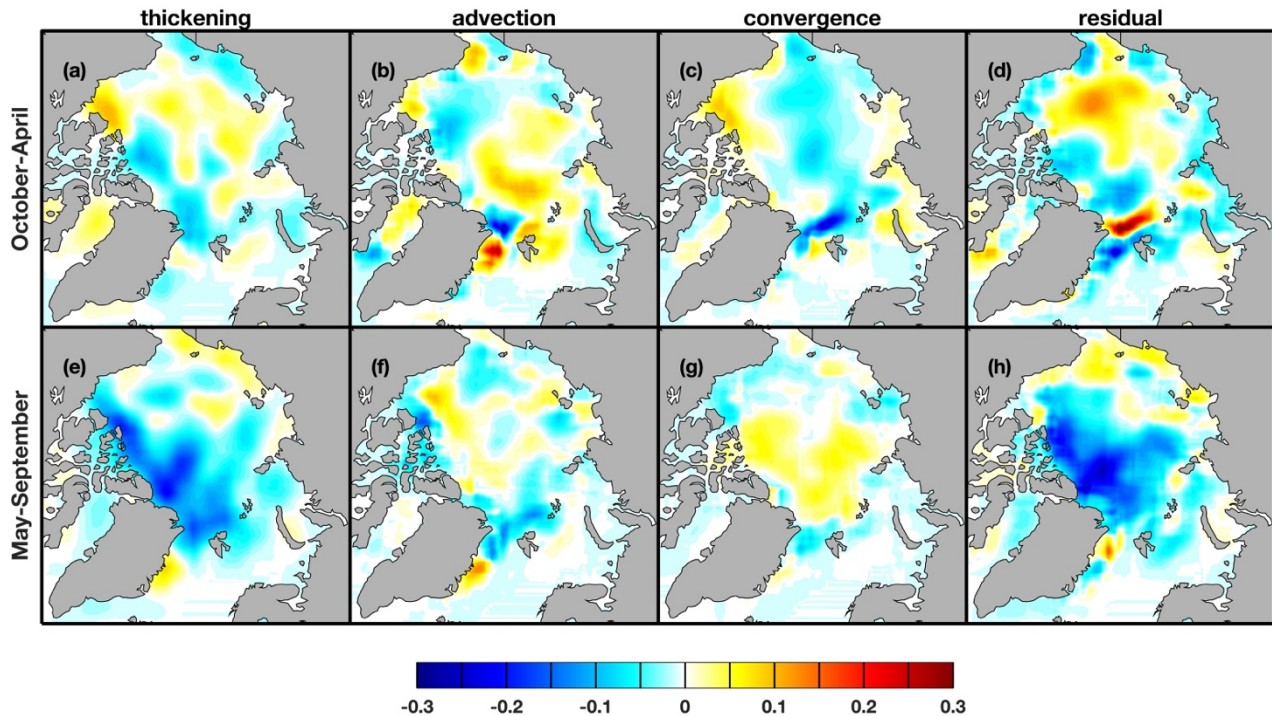

**Figure 3**: **Components of the Arctic sea ice thickness budget anomalies (unit: meters per month) from October 2010 to September 2011. The panel shows the Arctic sea ice thickness budget anomalies by subtracting the 6-year mean from each month.**








Figure 4: **Monthly mean arctic sea (a) Arctic Oscillation index (b) ice export anomalies at Fram Strait (unit: km3 month-1), (c) surface air temperature anomaly (unit: K) and (d) surface net heat flux anomaly (unit: W m-2).**



**Figure 5**: **Seasonal 2011 Arctic sea ice drift anomaly vector overlaid with the advection term ($-u \cdot \nabla H$; first row) and convergence term ($-H\nabla \cdot u$; second row). Winds anomalies at 10m (m/s) and sea level pressure anomalies (hPa) (bottom row).**

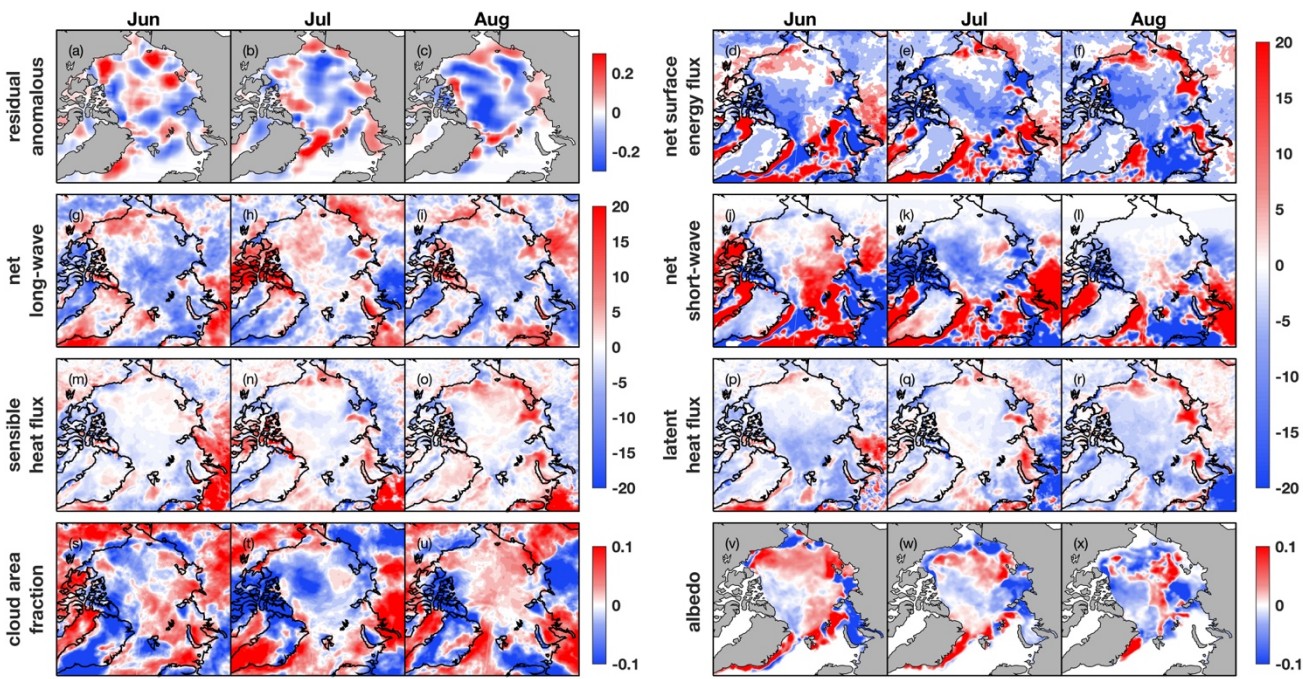


**Figure 6**: **The monthly thermodynamic residual anomalies (unit: meters per month) of the Arctic sea ice thickness budget in 2011 summer (a-c). Surface anomalies of net energy flux(d-f), net longwave radiation(g-i), net shortwave radiation(j-l), sensible heat fluxes(m-o) and latent heat flux (p-r) in 2011 summer (unit: W m⁻²). The cloud area fraction anomalies(s-u) and albedo**

**anomalies(v-x) are denoted in bottom row.**


**Table 1:** First- and multiyear ice (FYI/MYI) volume export through the Fram Strait in km$^3$ month$^{-1}$. The bold numbers indicate the anomalies are statistically significant.

| | | Jan | Feb | Mar | Apr | May | Jun | Jul | Aug | Sep | Oct | Nov | Dec |
|---|---|---|---|---|---|---|---|---|---|---|---|---|---|
| 2010 | FYI | - | - | - | - | - | - | - | - | - | -13 | -6 | -20 |
| | MYI | - | - | - | - | - | - | - | - | - | **-219** | -185 | -218 |
| | TOT | - | - | - | - | - | - | - | - | - | **-233±21** | -193±23 | -241±42 |
| 2011 | FYI | -36 | -4 | -132 | -71 | -71 | -36 | -25 | -15 | -1 | -7 | -14 | -15 |
| | MYI | **-201** | -29 | **-306** | -157 | **-205** | **-147** | **-90** | -26 | -26 | -142 | -161 | -274 |
| | TOT | **-238±15** | -34±64 | **-442±15** | -230±49 | -279±39 | -185±26 | -115±17 | -64±34 | -27±28 | -151±25 | -175±18 | -290±43 |
| 2012 | FYI | -4 | -32 | -69 | -147 | -166 | -103 | -144 | -91 | -29 | -36 | -27 | -28 |
| | MYI | -136 | -267 | -197 | -223 | -164 | -113 | -37 | -41 | -66 | -123 | -123 | -105 |
| | TOT | -137±25 | -300±38 | -267±19 | -372±21 | -334±21 | -218±19 | -187±25 | -131±42 | -95±21 | -160±13 | -149±16 | -134±23 |
| 2013 | FYI | -23 | -44 | -109 | -146 | -118 | -52 | -38 | -39 | -8 | -28 | -85 | -72 |
| | MYI | -59 | -64 | -101 | -69 | -72 | -90 | -69 | -60 | -18 | -198 | -281 | -117 |
| | TOT | -78±12 | -108±33 | -217±22 | -219±18 | -194±20 | -140±26 | -107±36 | -98±32 | -25±18 | -228±16 | -367±25 | -191±35 |
| 2014 | FYI | -28 | -65 | -128 | -229 | -132 | -105 | -64 | -85 | -26 | -6 | -10 | -55 |
| | MYI | -34 | -47 | -148 | -190 | -100 | -53 | -48 | -99 | -172 | -163 | -151 | -217 |
| | TOT | -61±21 | -114±14 | -282±14 | -425±13 | -232±13 | -161±20 | -112±28 | -184±8 | -198±11 | -170±5 | -162±17 | -275±26 |
| 2015 | FYI | -44 | -140 | -145 | -103 | -68 | -80 | -58 | 0 | -15 | -2 | -32 | -99 |
| | MYI | -76 | -211 | -188 | -199 | -99 | -154 | -53 | -11 | -92 | -74 | -157 | -136 |
| | TOT | -128±32 | -355±23 | -339±33 | -308±12 | -171±19 | -239±14 | -113±8 | -11±27 | -107±31 | -78±18 | -191±23 | -244±24 |
| 2016 | FYI | -69 | -117 | -80 | -161 | -126 | -89 | -56 | -71 | -5 | - | - | - |
| | MYI | -74 | -149 | -201 | -122 | -56 | -95 | -46 | -126 | -70 | - | - | - |
| | TOT | -145±24 | -267±11 | -287±23 | -289±8 | -196±9 | -194±6 | -113±26 | -198±55 | -75±13 | - | - | - |