# Peer review of "On the 2011 record low Arctic sea ice thickness: a combination of dynamic and thermodynamic anomalies"

_The Cryosphere, 2020_

## Referee Comment (RC1)

The Cryosphere Review:

tc-2020-359

On the 2011 record low Arctic sea ice thickness: a combination of dynamic and thermodynamic anomalies

Xuewei Li, Qinghua Yang, Lejiang Yu, Paul R. Holland, Chao Min, Longjiang Mu, Dake Chen

**Overview:**

The authors investigate the 2011 Arctic record low sea ice thickness minimum by examining satellite-derived ice thickness, area and volume, CMST modeled sea ice thickness and drift, ice transport through the Fram Strait, and the thermodynamic and dynamic processes which contributed to the record minimum.

Utilizing AWI and ESA CryoSat-2 ice thickness data, coupled with ice age data to differentiate between FYI and MYI, they examine the Arctic sea ice area, extent, and volume during the period of 2010-2020. They note the ice thickness minimum in Oct 2011, the third lowest observed ice area minimum in Fall 2012, and the decline of MYI from 2010-2012.

The CMST dataset generated from the MITgcm model which assimilated SMOS ice thickness and SSMIS ice concentration is used to analyze ice transport through the Fram Strait for the period of Oct 2010-September 2016. They find that declining trends in MYI volume in October 2010, January 2011 and March 2011 were associated with dynamic processes.

Dynamical and thermodynamic processes are evaluated during the study period to examine the role of ice thickening, advection, convergence, and residual (melting/freezing). These fields are examined for two periods: October 2010-April 2011, and May-September 2011. Anomalies of these terms are made by subtracting the 6-year mean for each month. They found a strong thinning of the ice cover along the Canadian Archipelago and portions of the central Arctic with enhanced melting (compared to 6-year mean) during the period of May – September 2011.

The Arctic Oscillation (data provided by NCEP) analysis in conjunction with the ERA5 reanalysis is used to examine the monthly variability between the AO, positive ice export anomalies through the Fram Strait, surface air temperature anomaly, and surface net heat flux anomaly. They find a positive AO from Feb-April 2011, a maximum ice export through the Fram Strait in March 2011 which coincided with strong air surface temperature and surface net heat flux anomalies in March 2011.

SLP anomalies from the ERA5 reanalysis for the period of October 2010-January 2011 showed a peak positive anomaly over southern Greenland, and a gradual divergence of sea ice along the eastern Arctic toward the northern CAA. Enhanced transpolar advection (Fig. 5a) showed the transport of sea ice from the Beaufort Sea along the coast toward the Fram Strait.

Lastly, they examine the radiative fluxes for June, July, and August 2011 to investigate the impact of cloud cover, albedos and net surface longwave radiation anomalies on the enhanced melt of primarily FYI during this period.

This is a well written paper which investigates the contributing factors which led to the record 2011 Arctic sea ice thickness minimum. A combination of satellite-derived products, the MITgcm-based CMST and the ERA5 reanalysis are used in this study. Figures and table are clear and easy to understand. All references appear to be correct. I recommend a minor revision. See comments below.

**General Comments:**

Use consistent use of shortwave and longwave (not short-wave, long-wave) throughout the paper. For example see lines: 202, 205, 215. Figure 6 caption uses correct form.

Lines 86-88: Gate positions at 82°N between 12°W and 20°E and 20°E between 80.5 and 82°N are defined. Which position(s) are included in Table 1?

Although not a major contributor to Arctic ice export, please comment on the role of the absence of ice arches (or bridges) in 2008-2009 and 2009-2010 (Ryan and Munchow, 2017) on the potential impact of ice export through the Fram Strait in those years.

**Technical Corrections:**

Line 74: Should be CryoSat-2

Line 82: What is the source of the sea ice concentration used?

Line 130: Do you mean CAA instead of CA?

Line 145: Same comment as above

Line 170: Include years October **2011**-April **2012**

Line 233: I assume you mean "net surface shortwave"?

Line 380: Include years of 2010, 2011 in caption.

Line 385: Why not show 4-month averages for all 3 plots shown: Oct-Jan, Feb-May, June-Sep? Any particular reason why Fig. 5g, 5h. 5i encompass a larger area than the two panels above?

Table 1: Include 6-year mean in bottom of table

Ryan, P. A. and A. Munchow (2017), Sea ice draft observations in Nares Strait from 2003 to 2012, J. Geophys. Res. Oceans, 122, 3057–3080, doi:10.1002/2016JC011966.

---

## Referee Comment (RC2)

First off, I disagree strongly with the statement that the ice thickness decline slowed down during CS2, there is no evidence of this, especially given the use of snow depth climatology (see Mallett et al. in TCD). All your statements about CS2 thickness variability should be stated with a caveat in that this assumes no interannual changes in snow depth, except as represented by first-year vs. multiyear ice. Thus, because this paper hinges on the 2011 minimum sea ice thickness as measured by CS2, then you will have to address the use of snow climatologies in these thickness estimates which are assimilated into your model. I'm also concerned about the lack of discussion on snow cover in general which plays an important role on thermodynamic ice growth as well as the timing of when bare ice and melt ponds form. Does the model not simulate any snow? The entire description of the modeling framework is too vague for this study. While references are given to the CMST model, you need to at least include some basic information such as resolution, atmospheric forcing data, etc. The entire methods section is weak and not suitable.

I also found the conclusions drawn often not supported by the data. In fact, the largest amount of ice in terms of area was not lost in 2011, and since you are further arguing that the ice was thinner, there is no way that you had the largest volume loss. The manuscript suffers from many of these types of inconsistencies, and vague statements without supporting evidence. Sadly I cannot recommend this paper for publication. It does not add any value to our understanding of processes in the Arctic, nor does it accurately portray the factors contributing to the "supposedly" anomalous thin ice in 2011.

Some specific line comments
Line 84: I don't believe you are using any method to track ice age, you are using a known data product and it should be stated as such

Line 89: you should also be aware if ice is advected towards the coast in the ice age product, the ice is lost (i.e. is transported onto land) so there will be a bias.

Line 90: you are not estimating anything here, instead you are using data from ERA5, unless this statement pertains to anomalies but then you need to specify how the anomalies are computed (i.e. relative to what reference period).

Line 113: I don't follow why you are only using a 6-year mean, you have a longer time-series and you should use it.

Line 125: I don't follow why you get enhanced winter melting in this region. I don't believe your residual term is entirely made up of thermodynamic processes, and I do not believe you have anomalous freshwater flux during this time. Where is the evidence for this? I think you are stretching your interpretations too far without the physics supporting these statements. What were your ocean and atmospheric temperatures in that region during that time?

Line 127: this is nothing new, divergence will result in thermodynamic ice growth that acts as a stabilizing feedback and there are many references the authors could cite about negative

feedbacks. Further, the thickness of the ice to start the growth season also plays an important role in this feedback process, and none of this is discussed. There are also two recent papers suggesting that the thermodynamic ice growth may be slowing, one by A. Petty (GRL) using climate model simulations and one by J. Stroeve (TC) using CS2 data in CICE.

Line 132: you cannot simply state that increased melt was driving by atmospheric temperature net surface heat flux and other variables. That is vague and uninformative. There are numerous factors that play a role in melt, including the timing of ice retreat and opening of leads/open water areas between the ice floes. You should at least try to quantify the relative contributions.

Line 145: I do not believe your assessment of enhanced ice export out of Fram Strait from October 2009 to January 2010 as it doesn't really match with my own calculations from at least 1 December through end of January. It is actually the second lowest amount of volume flux through the Fram Strait.

Line 177: More sea ice lost in 2011 than any other year? Again you haven't specified over what time-period this analysis is being done for, and it would be good for you to put this into the context also of total ice area lost.  If I compute the total ice extent lost between the maximum and minimum for each year, the maximum loss in total ice extent during summer is 2012, not 2011. And in fact 2011 is not even the second highest amount. If you are also arguing that you had thinner ice in 2011, then there is no way that you had more sea ice lost in 2011. Since this is an incorrect statement I didn't finish reading the rest. The entire paper is currently flawed, making statements that are not supported by the observations or the data used.

---

## Author Comment (AC1)

**Responses to referee #1**

Dear Reviewer:

We would like to thank you for the constructive comments to improve this manuscript. We overall agree with the points raised by the referee, and the points have been considered in revising the manuscript. In the following, our detailed responses are shown in italic.

Qinghua Yang

On behalf of all the co-authors

**General Comments:**

1) Use consistent use of shortwave and longwave (not short-wave, long-wave) throughout the paper. For example see lines: 202, 205, 215. Figure 6 caption uses correct form.

   *Response***:** *Thanks for the suggestion. We have now rephrased the texts in the revised.*

2) Lines 86-88: Gate positions at 82°N between 12°W and 20°E and 20°E between 80.5 and 82°N are defined. Which position(s) are included in Table 1?

   *Response***:** *Table 1 includes the sum of sea ice export volume through the gate positions at 82°N between 12°W and 20°E and 20°E between 80.5 and 82°N. We have now added a new figure to the Supplement of the revised version of our manuscript (as shown in the Figure S1 below). The thick green line in Figure S1 represents zonal and meridional sea ice gates to derive sea ice volume flux through the Fram Strait.*

[Figure]

Figure S1. Arctic sea ice thickness in September calculated from CMST (2011–2016). The thick green line represents zonal and meridional sea ice gates to derive sea ice volume flux through the Fram Strait. (f) The mean sea ice thickness is computed within the Arctic basin bounded by the gateways into the Pacific (Bering Strait), the CAA, and the Greenland (Fram Strait) and Barents Seas subdivided into maritime boundaries provided by NSIDC via MAISIE.

3) Although not a major contributor to Arctic ice export, please comment on the role of the absence of ice arches (or bridges) in 2008-2009 and 2009-2010 (Ryan and Munchow, 2017) on the potential impact of ice export through the Fram Strait in those years.

Ryan, P. A. and A. Munchow (2017), Sea ice draft observations in Nares Strait from 2003 to 2012, J. Geophys. Res. Oceans, 122, 3057–3080, doi:10.1002/2016JC011966.

*Response: The periods 2008-2009 and 2009-2010 actually are not spanned in our study. Nonetheless, we agree that the ice arches (or bridges) contribute to the ice export which is normally not well simulated in coarse sea ice models. We further addressed this potential effect in our manuscript:*

*"The sea ice arches over the channels and the fast-ice along the coast of the Arctic marginal seas may further add variability to the sea ice export through the Fram Strait (e.g., sea ice export study in Nares Strait by* Ryan and Münchow, 2017)*. This potential impact is not considered in current study and will be investigated in our further research."*

*Ryan, P. A. and Münchow, A.: Sea ice draft observations in Nares Strait from 2003 to 2012, J. Geophys. Res. Ocean., 122(4), 3057–3080, doi:10.1002/2016JC011966, 2017.*

**Technical Corrections:**

1) Line 74: Should be CryoSat-2

   **Response**: *Done.*

2) Line 82: What is the source of the sea ice concentration used?

   **Response**: *The sea ice concentration data uses the concentration that comes with the CS2 product.*

3) Line 130: Do you mean CAA instead of CA? Line 145: Same comment as above

   **Response**: *Thanks for the suggestion. We have now rephrased the texts in the revised: "In the regions where multi-year ice exists along the BS and CAA, a strong sea ice thinning is found, only very weakly offset by convergence and advection (Figure 3f,g)." and "The enhanced transpolar advection transported sea ice from the BS and CAA to the Fram Strait (Fig.5a)"*

4) Line 170: Include years October **2011**-April **2012**

   **Response**: *We have now reworded this sentence to: The total sea ice and multiyear sea ice fluxes through Fram Strait from October **2010**-April **2011** are 1611±229 km3 and 1315 km3, respectively (Table 1).*

5) Line 233: I assume you mean "net surface shortwave"?

*Response:* *We have now changed "net surface short" to "net surface shortwave" in the revised: In addition, the ice-albedo feedback also can modulate the net surface shortwave radiation.*

6) Line 380: Include years of 2010, 2011 in caption.

*Response:* *We have changed the caption, as suggested by the reviewer: "Figure 4: Monthly mean arctic sea (a) Arctic Oscillation index (b) ice export anomaly at Fram Strait (unit: km3 month-1; the positive anomaly represents more ice export), (c) surface air temperature anomaly (unit: K) and (d) surface net heat flux anomaly (unit: W m-2) from October 2010 to September 2011."*

7) Line 385: Why not show 4-month averages for all 3 plots shown: Oct-Jan, Feb-May, June-Sep? Any particular reason why Fig. 5g, 5h. 5i encompass a larger area than the two panels above?

*Response:* *The month division in Figure 5 is mainly based on the phase of AO index from October 2010 to September 2011. As shown in Figure 4a, AO has a continuous negative AO phase between Oct-Jan. In Feb-May, AO changes to a positive phase, while in June-Sep, AO returns to a negative phase.*

*The arrow in the top two panels represents the dynamic term of sea ice thickness budget and sea ice drift anomalies, so we need to plot the area with sea ice. Bottom panels show Winds and sea level pressure anomalies, so a larger area needs to be mapped to analyze the wind anomaly due to the pressure gradient.*

8) Table 1: Include 6-year mean in bottom of table

**Response:** We have included 6-year mean in bottom of table1.

Table 1: First- and multiyear ice (FYI/MYI) volume export through the Fram Strait in km3 month-1. The bold numbers indicate the anomalies are statistically significant.

| | | Jan | Feb | Mar | Apr | May | Jun | Jul | Aug | Sep | Oct | Nov | Dec |
|---|---|---|---|---|---|---|---|---|---|---|---|---|---|
| | FYI | - | - | - | - | - | - | - | - | - | -13 | -6 | -20 |
| 2010 | MYI | - | - | - | - | - | - | - | - | - | **-219** | -185 | -218 |
| | TOT | - | - | - | - | - | - | - | - | - | **-233±21** | -193±23 | -241±42 |
| | FYI | -36 | -4 | -132 | -71 | -71 | -36 | -25 | -15 | -1 | -7 | -14 | -15 |
| 2011 | MYI | **-201** | -29 | **-306** | -157 | **-205** | **-147** | **-90** | -46 | -26 | -142 | -161 | -274 |
| | TOT | **-238±15** | -34±64 | **-442±15** | -230±49 | -279±39 | -185±26 | -115±17 | -64±34 | -27±28 | -151±25 | -175±18 | -290±43 |
| | FYI | -4 | -32 | -69 | -147 | -166 | -103 | -144 | -91 | -29 | -36 | -27 | -28 |
| 2012 | MYI | -136 | -267 | -197 | -223 | -164 | -113 | -37 | -41 | -66 | -123 | -123 | -105 |
| | TOT | -137±25 | -300±38 | -267±19 | -372±21 | -334±21 | -218±19 | -187±25 | -131±42 | -95±21 | -160±13 | -149±16 | -134±23 |
| | FYI | -23 | -44 | -109 | -146 | -118 | -52 | -38 | -39 | -8 | -28 | -85 | -72 |
| 2013 | MYI | -59 | -64 | -101 | -69 | -72 | -90 | -69 | -60 | -18 | -198 | -281 | -117 |
| | TOT | -78±12 | -108±33 | -217±22 | -219±18 | -194±20 | -140±26 | -107±36 | -98±32 | -25±18 | -228±16 | -367±25 | -191±35 |
| | FYI | -28 | -65 | -128 | -229 | -132 | -105 | -64 | -85 | -26 | -6 | -10 | -55 |
| 2014 | MYI | -34 | -47 | -148 | -190 | -100 | -53 | -48 | -99 | -172 | -163 | -151 | -217 |
| | TOT | -61±21 | -114±14 | -282±14 | -425±13 | -232±13 | -161±20 | -112±28 | -184±8 | -198±11 | -170±5 | -162±17 | -275±26 |
| | FYI | -44 | -140 | -145 | -103 | -68 | -80 | -58 | 0 | -15 | -2 | -32 | -99 |
| 2015 | MYI | -76 | -211 | -188 | -199 | -99 | -154 | -53 | -11 | -92 | -74 | -157 | -136 |
| | TOT | -128±32 | -355±23 | -339±33 | -308±12 | -171±19 | -239±14 | -113±8 | -11±27 | -107±31 | -78±18 | -191±23 | -244±24 |
| | FYI | -69 | -117 | -80 | -161 | -126 | -89 | -56 | -71 | -5 | - | - | - |
| 2016 | MYI | -74 | -149 | -201 | -122 | -56 | -95 | -46 | -126 | -70 | - | - | - |
| | TOT | -145±24 | -267±11 | -287±23 | -289±8 | -196±9 | -194±6 | -113±26 | -198±55 | -75±13 | - | - | - |
| | FYI | -34 | -67 | -110 | -143 | -113 | -77 | -64 | -50 | -14 | -15 | -29 | -48 |
| mean | MYI | -97 | -128 | -190 | -160 | -116 | -109 | -57 | -64 | -74 | -153 | -176 | -178 |
| | TOT | -131±21 | -196±30 | -306±19 | -307±20 | -234±20 | -190±18 | -125±23 | -114±36 | -88±21 | -170±15 | -206±18 | -229±34 |

---

## Author Comment (AC2)

**Responses to referee #2**

Dear Reviewer:

You offers strong criticism of the paper and does not review all of it. We believe we can answer all of the criticisms stated.

The paper is not unusual in that it considers interannual variability in satellite altimetry of ice thickness, in the context of climatic forcing. These kinds of studies are widely published(Kwok, 2018; Kwok and Cunningham, 2015; Tilling et al., 2015). While there are of course important caveats to the use of these data, e.g. snow loading, these published studies demonstrate that it is widely accepted that satellite altimetry may be used to study variability in sea ice thickness.

We think the satellite data do show a real signal of anomalously thin ice, that deserves investigation. We accept of course that there are important uncertainties in the data, particularly caused by interannual variability in the snow loading. The revised paper will more fully present these uncertainties and consider them in the discussion of the conclusions.

It should be noted, as suggested by the reviewer, we agree that it is not rigorous to discuss trends and minima of sea ice thickness with the limited data. The value of the paper is more in revealing the underlying effects that cause the ice thickness/volume anomaly in 2011. So, we removed the texts related to the minimum and trend of sea ice thickness in the manuscript.

Qinghua Yang

On behalf of all the co-authors

**General Comments:**

1) First off, I disagree strongly with the statement that the ice thickness decline slowed down during CS2, there is no evidence of this, especially given the use of snow depth climatology (see Mallett et al. in TCD).

*Response: As mentioned by Kwok (2018) : "In the satellite record, the five ICESat years seem to have captured the steep declines in thickness (especially the sharp decrease in thickness after the record setting years of 2005 and 2007); the thinning seems to have slowed in theCS-2 years." However, as suggested by the reviewer, we agree that it is not rigorous to discuss the slow down ice thickness trends with the limited data and CS2 uncertainties of sea ice thickness. In addition to that, this is not the central claim of our paper. Our paper mainly focusses on the 2011 anomalies, not the change in thickness trend. The two are linked of course, but our work is really about 2011. So, we removed the text related to the trend of sea ice thickness in the manuscript.*

2) All your statements about CS2 thickness variability should be stated with a caveat in that this assumes no interannual changes in snow depth, except as represented by first-year vs. multiyear ice. Thus, because this paper hinges on the 2011 minimum sea ice thickness as measured by CS2, then you will have to address the use of snow climatologies in these thickness estimates which are assimilated into your model.
I'm also concerned about the lack of discussion on snow cover in general which plays an important role on thermodynamic ice growth as well as the timing of when bare ice and melt ponds form.

*Response: We agree with the reviewer that the lack of snow data and the uncertain radar penetration into snow is a significant weakness of any product based on CS2. The observational CS2 uncertainties of sea ice thickness contain contributions that are associated with speckle noise, sea-surface height estimation, snow depth and densities of ice and snow (Ricker et al., 2014). CS2 data have relatively large errors over the thin ice area, while SMOS has smaller error, and vice versa for the thick ice area (Ricker et al., 2017). So we replaced the data with CS2SMOS for consistency to compare the daily behavior of sea ice*

*thickness and volume from October 2010 through April 2020 and calculated the uncertainties in SIT as shown in the Figure S1 below. About the contribution of snow on CS2 uncertainties of sea ice thickness, the CS2 use the snow climatologist in thickness estimates. As shown by Fig.5 in Mallett et al.(2020), there was no obvious snow anomaly contributions to sea ice thickness in the Central Arctic in October 2011.*

*We also believe that the sampled radar signal is real, and that we broadly believe it's interpretation as an ice thickness trend, as shown by Mallett et al.(2020). We would also note in general that these ice thickness data have been used in many high-profile previous studies(Kwok, 2018; Min et al., 2019; Ricker et al., 2017b; Tilling et al., 2015), so the snow thickness issues do not in general prevent us from studying ice thickness anomalies.*

*Following your comments, we added a discussion on the uncertainty of CS2SMOS to the manuscript: The combined Cryosat-2 and SMOS satellite data (CS2SMOS) data from AWI is also used to compare the daily behavior of sea ice thickness and volume from October 2010 through April 2020 and calculated the uncertainties in SIT (Ricker et al., 2017a). In addition, Systematic errors as associated with the lack of interannual variability in the Warren snow climatology (Warren et al., 1999) or due to variable snow penetration will increase the uncertainty of altimetry-based thicknesses (Ricker et al., 2014). The snow data with more realistic variability and trends has wide implications for thickness variability in marginal seas (Mallett et al., 2020). The SMOS retrieval can contribute valuable information, especially in regions with uncertain snow depth estimates.*

[Figure]

Figure S1: Daily behavior of sea ice thickness and volume based on CS2SMOS dataset from October 2010 through April 2020. (a) Mean sea ice thickness within area of actual ice coverage. (b) Total(black),first-year(blue) and multiyear (red) sea ice volumes within Arctic basin. The mean sea ice thickness is computed within the area of actual ice coverage bounded by the gateways into the Pacific (Bering Strait), the Canadian Arctic Archipelago, and the Greenland (Fram Strait) and Barents Seas.

3) Does the model not simulate any snow? The entire description of the modeling framework is too vague for this study. While references are given to the CMST model, you need to at least include some basic information such as resolution, atmospheric forcing data, etc. The entire methods section is weak and not suitable.

*Response: Regarding the description of the model, note this was deliberate as we were citing an earlier paper, but of course it is fine to add more details here. So we refined this description as: We apply this methodology to a well-validated sea ice thickness and drift dataset (the Combined Model and Satellite Thickness data, CMST), which was generated by the MITgcm ice-ocean model with CryoSat-2, SMOS sea ice thickness and SSMIS sea ice concentration assimilated (Mu et al., 2018). Both the ocean and sea ice model are discretized on an Arakawa C grid*

*with a grid spacing of 18 km. In the ocean model, there are 50 unevenly spaced layers in the vertical direction. The atmospheric ensemble forecasts of the United Kingdom Met Office (UKMO) Ensemble Prediction System (EPS; Bowler et al., 2008) available in the THORPEX Interactive Grand Global Ensemble (TIGGE) are used to drive the ice-ocean model. The thermodynamics of sea ice use a one-layer, zero-heat capacity formulation (Semtner Jr, 1976; Parkinson &Washington, 1979) and the snow thickness is a prognostic variable following Zhang et al. (1998). The CMST thickness data cover both the cold seasons and the melting seasons for the period of October 2010 to December 2016. The CMST has been already quantitatively evaluated against observations by previous studies (Mu et al., 2018; Min et al., 2019; Min et al., 2021), demonstrating an accurate performance in simulating the real sea ice drift and thickness.*

4) I also found the conclusions drawn often not supported by the data. In fact, the largest amount of ice in terms of area was not lost in 2011, and since you are further arguing that the ice was thinner, there is no way that you had the largest volume loss.

*__Response__: Although the loss of sea ice area in 2011 was not the largest, the loss of sea ice thickness from October 2010 to September 2011 was also one of the important factors affecting the loss of sea ice volume. We have examined the whole satellite record, which includes pan-Arctic SIT snapshots from ICESAT (2003-2008) and CryoSat-2 (2011–2020) satellite datasets. We have now added a new figure to the Supplement of the revised version of our manuscript (as shown in the Figure S2 below) and compared the sea ice volume, area and mean sea ice thickness based on the ICESat (2003–2008) and CryoSat-2 (2011–2020) satellite datasets. That clearly shows that the volume loss and thickness loss are both largest.*

*This conclusion is consistent with several published studies. We argue that the fact that* Kwok (2018) *has a paper published discussing Arctic sea ice volume in 2011 hit the lowest record from 0ct. to Nov. between 2003 and 2018 in the same Arctic basin shows that i) the satellite data are considered worthy of studying and ii) this individual event is worthy of studying. As mentioned by* Tilling et al.(2015) *in Nature Geoscience : "It is notable, for example, that the record minimum Arctic sea ice extent of September 2012 was accompanied by thicker autumn ice in this region*

*than in previous years, demonstrating that decreasing ice extent does not necessarily result in a proportionate decrease in ice volume". Although only the data from 2010 to 2014 were available, Tilling et al. (2015) showed that the total loss of sea ice volume from autumn 2010 ($9.03*10^3 km^3$) to autumn 2011 ($7.86*10^3 km^3$) was the largest in the five years (2010-2014) (as shown in the Table S1 below). As suggested by reviewer, the total ice extent lost in 2011 was not the largest, but the ice volume loss was. This is not inconsistent; it just shows that there was an abnormal loss of sea ice thickness from October 2010 to September 2011. We disagree that our study does not add value, and we believe that there was thinner ice in 2011, and that the factors are discussed. We are happy to address any specific issues that the reviewer would like to raise that concern these points.*

[Figure]

Figure S2: Interannual changes in sea ice volume, area and thickness based on the ICESat (2003–2008) and CryoSat-2 (2011–2020) satellite datasets. (a) Mean sea ice thickness within area of actual ice coverage. (b) Total sea ice area (cumulative area of actual ice coverage) within Arctic basin. (c) Total(black),first-year(blue) and multiyear (red) sea ice volumes within Arctic basin. Arctic basin volume and area is computed within the bounded by the gateways into the Pacific (Bering Strait), the Canadian Arctic Archipelago, and the Greenland (Fram Strait) and Barents Seas.

Table S1. Table 1 in Tilling et al. (2015).

**Table 1 | Average CryoSat-2 Arctic sea ice volume (10³ km³) for autumn (October–November) 2010–2014 and spring (March–April) 2011–2014.**

| Year | Volume (MYI) | | Volume (FYI) | | Volume (total) | |
|------|--------------|---|--------------|---|----------------|---|
| | Autumn | Spring | Autumn | Spring | Autumn | Spring |
| 2010–2011 | 5.34 ± 0.69 | 7.64 ± 0.94 | 3.69 ± 0.59 | 17.99 ± 2.44 | 9.03 ± 1.28 | 25.63 ± 3.37 |
| 2011–2012 | 3.75 ± 0.56 | 5.72 ± 0.71 | 4.11 ± 0.63 | 19.57 ± 2.66 | 7.86 ± 1.19 | 25.29 ± 3.36 |
| 2012–2013 | 3.70 ± 0.48 | 6.23 ± 0.80 | 4.05 ± 0.62 | 18.20 ± 2.53 | 7.75 ± 1.10 | 24.43 ± 3.32 |
| 2013–2014 | 6.95 ± 0.82 | 9.63 ± 1.12 | 3.99 ± 0.61 | 16.96 ± 2.29 | 10.94 ± 1.43 | 26.59 ± 3.41 |
| 2014–2015 | 6.18 ± 0.73 | - | 4.08 ± 0.62 | - | 10.26 ± 1.34 | - |

To estimate uncertainties in monthly sea ice volume, we account for uncertainties in the sea ice density, snow density, snow depth, sea ice area, sea ice concentration, and for spatial variations in the measurement of sea ice freeboard. The autumn and spring uncertainties are the averaged uncertainties of their corresponding months.

5) The manuscript suffers from many of these types of inconsistencies, and vague statements without supporting evidence. Sadly I cannot recommend this paper for publication. It does not add any value to our understanding of processes in the Arctic, nor does it accurately portray the factors contributing to the "supposedly" anomalous thin ice in 2011.

*Response: We believe the manuscript is not inconsistent, and that we are happy to address any examples of inconsistency that the reviewers raise. We believe there is plenty of evidence in our paper and in the literature that the 2011 anomaly is real. The dynamic and thermodynamic processes leading to the dramatic sea ice thickness loss are described in Sect. 3. We have performed a detailed investigation of the 2011 anomaly, and we think this clearly does add value to our understanding of Arctic processes.*

**Some specific line comments**

1) Line 84: I don't believe you are using any method to track ice age, you are using a known data product and it should be stated as such.

*Response: We refined this description as: We also use the weekly sea ice age for the Arctic Ocean introduced by Fowler et al. (2003) and described further by Maslanik et al. (2007), Tschudi et al. (2010), Stroeve et al. (2011) and Tschudi et al. (2020). In the Fowler et al. (2003) approach, the method used to estimate sea ice age involves Lagrangian tracking of sea ice from week-to-week using gridded ice motion vectors.*

2) Line 89: you should also be aware if ice is advected towards the coast in the ice age

product, the ice is lost (i.e. is transported onto land) so there will be a bias.

*Response: Following your advice, we changed this sentence as: Note that motions are not retrieved near coasts, because motion retrievals near the coast are unreliable due to the effects of mixed land and ice/ocean grid cells (Tschudi et al., 2019). Thus, the sum of FYI and MYI is slightly less than the total amount of ice.*

3) Line 90: you are not estimating anything here, instead you are using data from ERA5, unless this statement pertains to anomalies but then you need to specify how the anomalies are computed (i.e. relative to what reference period).

*Response: We agree that our explanation was not sufficient, so we refined this description as: In this study, to quantify the thermodynamic impact on the ice thickness budget, we estimate sea level pressure (SLP), 10 m wind speed, surface radiation fluxes, and albedo anomalies by subtracting the 6-year mean (from October 2010 to September 2016) for each month, derived from monthly ERA5 atmospheric reanalysis data from the European Centre for Medium-Range Weather Forecasts (ECMWF)(Copernicus Climate Change Service (C3S), 2017; Hersbach et al., 2020) .*

4) Line 113: I don't follow why you are only using a 6-year mean, you have a longer time-series and you should use it.

*Response: Because the CMST sea ice thickness and drift dataset covers both the cold seasons and the melting seasons only for the period of October 2010 to December 2016.*

5) Line 125: I don't follow why you get enhanced winter melting in this region. I don't believe your residual term is entirely made up of thermodynamic processes, and I do not believe you have anomalous freshwater flux during this time. Where is the evidence for this? I think you are stretching your interpretations too far without the physics supporting these statements. What were your ocean and atmospheric temperatures in that region during that time?

*Response: The combination of figures 2 and 3 in the paper (Fig2d and Fig3d) show that north-east of Greenland, in the climatology there is freezing to the north, and then melting to the south, as the ice is advected south into a less cold climate. In*

*2011 this climate is cooler, so freezing extends further south, but the ice advection is faster, so there is more melting south of the freezing line. This leads to a 'dipole' in thermodynamic anomalies, in figure 3d.*

*As shown in Fig S3, the mean air temperature and sea surface temperature from October 2010 to April 2011 show this cold climatic anomaly north-east of Greenland.*

[Figure]

Figure S3: Mean sea surface temperature (SST) anomalies and mean air temperature at 2m (t2m) anomalies from October 2010 to April 2011.

6) Line 127: this is nothing new, divergence will result in thermodynamic ice growth that acts as a stabilizing feedback and there are many references the authors could cite about negative feedbacks. Further, the thickness of the ice to start the growth season also plays an important role in this feedback process, and none of this is discussed. There are also two recent papers suggesting that the thermodynamic ice growth may be slowing, one by A. Petty (GRL) using climate model simulations and one by J. Stroeve (TC) using CS2 data in CICE.

*Response: We are simply claiming that we found that analyzing the budget anomalies in the context of the mean budget (figures 2 and 3) is helpful as it clarifies exactly how the atmospheric wind and thermodynamic forcing modifies the ice growth here. Compensation between dynamics and thermodynamics is of course expected, but we believe it is extremely informative to see the exact patterns and rates of thermodynamic and dynamic contributions to overall anomalies. Following the comments, we have added the discussion on the feedback between*

*dynamics and thermodynamics: The thickness of the ice to start the growth season also plays an important role in this feedback process. Stroeve et al. (2018) and Petty et al. (2018) highlighted the importance of the negative winter growth feedback mechanism—thinner ice grows faster than thicker ice due to its decreased insulation. Thus, although the summer sea ice in 2011 is rapidly declining, the negative feedbacks over winter allow for recovery following low summer's sea ice thickness.*

7) Line 132: you cannot simply state that increased melt was driving by atmospheric temperature net surface heat flux and other variables. That is vague and uninformative. There are numerous factors that play a role in melt, including the timing of ice retreat and opening of leads/open water areas between the ice floes. You should at least try to quantify the relative contributions.

*__Response:__ The sea ice thickness budget contributions caused by thermodynamic processes in response to the driving climatic factors are described in Sect. 3.3. It is unfortunate that the reviewer didn't read section 3.*

8) Line 145: I do not believe your assessment of enhanced ice export out of Fram Strait from October 2009 to January 2010 as it doesn't really match with my own calculations from at least 1 December through end of January. It is actually the second lowest amount of volume flux through the Fram Strait.

*__Response:__ First, we did not calculate the sea ice export volume from the Fram Strait from October 2009 to January 2010 in this manuscript. The period we calculated is from October 2010 to September 2016, which is indicated in line 145. Second, in terms of the sea ice export volume from the Fram Strait during October 2010 to September 2011,* Min et al. (2019) *showed the same result by estimating the seasonal and interannual variations of Arctic sea ice volume flux through the Fram Strait from September 2010 to December 2016. Ricker et al. (2018) showed a similar result using OSI SAF ice drift in their Table 2. Although we used different datasets, the sea ice export volume in Jan-Mar 2011 also show statistically significant anomalies (as shown in the table below).*

Table S2. Table 2 in Ricker et al. (2018)).

**Table 2.** Monthly Arctic sea ice volume export through the Fram Strait in km$^3$ month$^{-1}$ computed with OSI SAF ice drift. Maximum and minimum values are denoted in italic and bold, respectively.

|           | Oct   | Nov   | Dec   | Jan   | Feb   | Mar   | Apr   | Mean  |
|-----------|-------|-------|-------|-------|-------|-------|-------|-------|
| 2010–2011 | –     | −227  | −275  | −267  | **−21** | *−540* | −279  | −268  |
| 2011–2012 | −164  | −214  | −354  | −129  | −381  | −379  | −487  | −301  |
| 2012–2013 | −203  | −182  | −187  | −103  | −163  | −299  | −318  | −208  |
| 2013–2014 | −215  | −400  | −231  | −78   | −195  | −345  | −452  | −274  |
| 2014–2015 | −200  | −165  | −373  | −160  | −425  | −429  | −354  | −301  |
| 2015–2016 | −52   | −261  | −275  | −177  | −352  | −348  | −310  | −254  |

9) Line 177: More sea ice lost in 2011 than any other year? Again you haven't specified over what time-period this analysis is being done for, and it would be good for you to put this into the context also of total ice area lost. If I compute the total ice extent lost between the maximum and minimum for each year, the maximum loss in total ice extent during summer is 2012, not 2011. And in fact 2011 is not even the second highest amount. If you are also arguing that you had thinner ice in 2011, then there is no way that you had more sea ice lost in 2011. Since this is an incorrect statement I didn't finish reading the rest. The entire paper is currently flawed, making statements that are not supported by the observations or the data used

*Response**: Thanks for this comment. We agree that our explanation could have been more specific, so we refined this description as: Compared with the 6-year mean (from May 2011 to September 2016), the sea ice thickness budget from May to September 2011 shows a negative anomaly, indicating that the loss of sea ice thickness increases during the season of sea ice retreat.*

*As for the problem of volume loss of sea ice, we have replied in the second point of General Comments, that indeed 2011 did have the most ice loss, according to us and to other authors.*

i) *Although the loss of sea ice area in 2011 was not the largest, the loss of sea ice thickness from October 2010 to September 2011 was also one of the important factors affecting the loss of sea ice volume. As mentioned by Tilling et al.(2015) in Nature Geoscience : "It is notable, for example, that the record minimum Arctic sea ice extent of September 2012 was accompanied by thicker autumn ice in this region than in previous years, demonstrating that decreasing ice extent does not necessarily result in a proportionate decrease in ice volume".*

ii) *Kwok (2018) has a paper published discussing Arctic sea ice volume in 2011*

*hit the lowest record from 0ct. to Nov. between 2003 and 2018 in the same Arctic basin. Although only the data from 2010 to 2014 were available, Tilling et al. (2015) showed that the total loss of sea ice volume from autumn 2010 (9.03\*10³km³) to autumn 2011 (7.86\*10³km³) was the largest in the five years (2010-2014) (as shown in the Table S1 below).*

Table S1. Table 1 in Tilling et al. (2015).

**Table 1 | Average CryoSat-2 Arctic sea ice volume ($10^3$ km³) for autumn (October–November) 2010–2014 and spring (March–April) 2011–2014.**

| Year | Volume (MYI) | | Volume (FYI) | | Volume (total) | |
|---|---|---|---|---|---|---|
| | Autumn | Spring | Autumn | Spring | Autumn | Spring |
| 2010–2011 | 5.34 ± 0.69 | 7.64 ± 0.94 | 3.69 ± 0.59 | 17.99 ± 2.44 | 9.03 ± 1.28 | 25.63 ± 3.37 |
| 2011–2012 | 3.75 ± 0.56 | 5.72 ± 0.71 | 4.11 ± 0.63 | 19.57 ± 2.66 | 7.86 ± 1.19 | 25.29 ± 3.36 |
| 2012–2013 | 3.70 ± 0.48 | 6.23 ± 0.80 | 4.05 ± 0.62 | 18.20 ± 2.53 | 7.75 ± 1.10 | 24.43 ± 3.32 |
| 2013–2014 | 6.95 ± 0.82 | 9.63 ± 1.12 | 3.99 ± 0.61 | 16.96 ± 2.29 | 10.94 ± 1.43 | 26.59 ± 3.41 |
| 2014–2015 | 6.18 ± 0.73 | - | 4.08 ± 0.62 | - | 10.26 ± 1.34 | - |

To estimate uncertainties in monthly sea ice volume, we account for uncertainties in the sea ice density, snow density, snow depth, sea ice area, sea ice concentration, and for spatial variations in the measurement of sea ice freeboard. The autumn and spring uncertainties are the averaged uncertainties of their corresponding months.

*References:*

Copernicus Climate Change Service (C3S): ERA5: Fifth generation of ECMWF atmospheric reanalyses of the global climate, Copernicus Clim. Chang. Serv. Clim. Data Store (CDS), accessed 2018-05-04, 2017.

Hersbach, H., Bell, B., Berrisford, P., Hirahara, S., Horányi, A., Muñoz-Sabater, J., Nicolas, J., Peubey, C., Radu, R., Schepers, D., Simmons, A., Soci, C., Abdalla, S., Abellan, X., Balsamo, G., Bechtold, P., Biavati, G., Bidlot, J., Bonavita, M., De Chiara, G., Dahlgren, P., Dee, D., Diamantakis, M., Dragani, R., Flemming, J., Forbes, R., Fuentes, M., Geer, A., Haimberger, L., Healy, S., Hogan, R. J., Hólm, E., Janisková, M., Keeley, S., Laloyaux, P., Lopez, P., Lupu, C., Radnoti, G., de Rosnay, P., Rozum, I., Vamborg, F., Villaume, S. and Thépaut, J. N.: The ERA5 global reanalysis, Q. J. R. Meteorol. Soc., 146(730), 1999–2049, doi:10.1002/qj.3803, 2020.

Kwok, R.: Arctic sea ice thickness, volume, and multiyear ice coverage: Losses and coupled variability (1958-2018), Environ. Res. Lett., 13(10), 105005, doi:10.1088/1748-9326/aae3ec, 2018.

Kwok, R. and Cunningham, G. F.: Variability of arctic sea ice thickness and volume from CryoSat-2, Philos. Trans. R. Soc. A Math. Phys. Eng. Sci., 373(2045), 20140157, doi:10.1098/rsta.2014.0157, 2015.

Mallett, R., Stroeve, J., Tsamados, M., Landy, J., Willatt, R., Nandan, V. and Liston, G.: Faster decline and higher variability in the sea ice thickness of the marginal Arctic seas, Cryosph. Discuss., (October), 1–31, doi:10.5194/tc-2020-282, 2020.

Min, C., Mu, L., Yang, Q., Ricker, R., Shi, Q., Han, B., Wu, R. and Liu, J.: Sea ice export through the Fram Strait derived from a combined model and satellite data set, Cryosph. Discuss., 13(12), 3209–3224, doi:10.5194/tc-2019-157, 2019.

Petty, A. A., Holland, M. M., Bailey, D. A. and Kurtz, N. T.: Warm Arctic, Increased Winter Sea Ice Growth?, Geophys. Res. Lett., 45(23), 12,922-12,930, doi:10.1029/2018GL079223, 2018.

Ricker, R., Hendricks, S., Kaleschke, L., Tian-Kunze, X., King, J. and Haas, C.: A weekly Arctic sea-ice thickness data record from merged CryoSat-2 and SMOS satellite data, Cryosphere, 11(4), 1607–1623, doi:10.5194/tc-11-1607-2017, 2017a.

Ricker, R., Hendricks, S., Girard-Ardhuin, F., Kaleschke, L., Lique, C., Tian-Kunze, X., Nicolaus, M. and Krumpen, T.: Satellite-observed drop of Arctic sea ice growth in winter 2015–2016, Geophys. Res. Lett., 44(7), 3236–3245, doi:10.1002/2016GL072244, 2017b.

Ricker, R., Girard-Ardhuin, F., Krumpen, T. and Lique, C.: Satellite-derived sea ice export and its impact on Arctic ice mass balance, Cryosphere, 12(9), 3017–3032, doi:10.5194/tc-12-3017-2018, 2018.

Stroeve, J. C., Schroder, D., Tsamados, M. and Feltham, D.: Warm winter, thin ice?, Cryosphere, 12(5), 1791–1809, doi:10.5194/tc-12-1791-2018, 2018.

Tilling, R. L., Ridout, A., Shepherd, A. and Wingham, D. J.: Increased Arctic sea ice volume after anomalously low melting in 2013, Nat. Geosci., 8(8), 643–646, doi:10.1038/ngeo2489, 2015.

Tschudi, M. A., Meier, W. N. and Stewart, J. S.: An enhancement to sea ice motion and age products, Cryosph. Discuss., 1–29, doi:10.5194/tc-2019-40, 2019.

---

## Author Comment (AC3)

**Responses to Robbie Mallett**

Dear Robbie Mallett:

We would like to thank you for the constructive comments to improve this manuscript. We overall agree with the points raised, and the points have been considered in revising the manuscript. In the following, our detailed responses are shown in italic.

Qinghua Yang

On behalf of all the co-authors

**General Comments:**

1)  I read this paper with great interest as a user of CryoSat-2 data, and have a couple of thoughts regarding the authors' use of this data in the spirit of TC discussion.

    It seems that a headline result of this study is that "a sea ice thickness record minimum is confirmed occurring in autumn 2011" (I hope this is fair to say given that it's the second sentence of the abstract and first of the Summary/Discussion). I think to fully make this claim there should perhaps be a deeper consideration of the nature of this metric and the uncertainties in altimetry-derived SIT (particularly over thin ice).

    To state the obvious, SIT is a local property of sea ice, whereas extent and volume are global properties of the ice pack. The impact of this is that mean SIT is sensitive to the area over which it's averaged (unlike the other two metrics). The approach in this paper is to just average over the "area of actual ice coverage" (L37). The decision to average over this area this has many implications.

    For instance, are the authors including the sub-Arctic seas like the Baltic and Okhotsk Seas and Baffin and Hudson Bays? I believe the AWI product includes SIT

values for these. If the SIT of these regions contributes to the 'mean SIT' statistic, then how relevant is their interpretation of the 2011 minimum in terms of the dynamic/thermodynamic budget which is only produced for the Arctic Ocean (and also only with reference to 2011-16).

*Response: Thanks for this comment. We found the previous context may not sufficiently describe the region where we compute the mean sea ice thickness. We have now added in more detail: "The mean sea ice thickness is computed within the area of actual ice coverage bounded by the gateways into the Pacific (Bering Strait), the Canadian Arctic Archipelago, and the Greenland (Fram Strait) and Barents Seas (Kwok, 2018; Kwok and Cunningham, 2015)." The region doesn't include the sub-Arctic seas. We have added a new figure to the Supplement of the revised version of our manuscript (as shown in the Figure S1f below).*

*About the sub-Arctic seas contributes to the 'mean SIT' statistic, we compared the mean SIT calculated by the whole-Arctic and only for the Arctic Ocean as shown in the Figure S2 below. About the uncertainties in altimetry-derived SIT, we also calculated the uncertainties in CS2SMOS. The mean sea ice thickness in both results in October 2011 was strongly anomalous.*

*We agree that the SIT is a local property of sea ice and sensitive to the area of actual ice coverage within the boundaries in figure S1. If the SIT is calculated using fixed area instead of the actual area covered by sea ice, then the information reflected by SIT is actually the same as the volume. The influence of sea ice is not just about how much ice volume there is. It is about how that ice is distributed. In 2011 the CS2 data suggest that the ice is unusually thinly distributed as shown in the Figure S1 below, and so that justifies a closer examination of what happened in that year.*

[Figure]

**Figure S1.** Arctic sea ice thickness in September calculated from CMST (2011–2016). The thick green line represents zonal and meridional sea ice gates to derive sea ice volume flux through the Fram Strait. (f) The mean sea ice thickness is computed within the Arctic basin bounded by the gateways into the Pacific (Bering Strait), the CAA, and the Greenland (Fram Strait) and Barents Seas subdivided into maritime boundaries provided by NSIDC via MAISIE.

[Figure]

Figure S2: Daily behavior of sea ice thickness and volume based on CS2SMOS dataset from October 2010 through April 2020. (a) Mean sea ice thickness within area of whole-Arctic mean SIT including the sub-Arctic sea. (b) Mean sea ice thickness within area only for the Arctic Ocean. The mean sea ice thickness is computed within the area of actual ice coverage bounded by the gateways into the Pacific (Bering Strait), the Canadian Arctic Archipelago, and the Greenland (Fram Strait) and Barents Seas.

2) Next the authors should probably acknowledge that CryoSat-2 doesn't do a good job of retrieving the thickness of thin ice (<0.5m; see Ricker et al., 2017). This is because this ice protrudes above the waterline by less than 5cm, and even less with snow cover. But here I think the authors are averaging over quite a lot of thin ice to generate their statistic. The merged CS2-SMOS product was developed with this limitation in mind, and (in my opinion) should be the product of choice for this calculation. I particularly think this because the CMST model assimilated this product, so it should perhaps be used for consistency's sake anyway. A related issue is that CS2 simply can't measure ice below a certain thickness. By

just taking the average in places where it can measure, the authors are likely biasing their mean SIT statistic high. The size of this bias will depend on the extent of sea ice with unmeasurably low freeboard. I think they should state what fraction of the total sea ice area (as measured by a scatterometer or radiometer) is covered by the altimetry data under consideration. It's possible that this fraction is very high and my concern isn't warrented, but I think it is relevant.

*Response: Thanks for this comment. The observational CS2 uncertainties of sea ice thickness contain contributions that are associated with speckle noise, sea-surface height estimation, snow depth and densities of ice and snow (Ricker et al., 2014). CS2 data have relatively large errors over the thin ice area, while SMOS has smaller error, and vice versa for the thick ice area (Ricker et al., 2017). As suggested by the reviewer, we replaced the data with CS2SMOS to compare the daily behavior of sea ice thickness and volume from October 2010 through April 2020 and calculated the uncertainties as shown in the Figure S3 below. The retrieval method is based on the evaluation of surface brightness temperature, as measured by the Soil Moisture and Ocean Salinity (SMOS) satellite. The uncertainties over thin ice are significantly smaller than for the altimetry- based retrievals. The SMOS retrieval can contribute valuable information, especially in regions with unmeasurably low freeboard. The complementary character between CS2 and SMOS made up the bias caused by unmeasurably low freeboard. At the same time, the results of ICESAT and CryoSat-2 data are put into Supplementary of the revised version of our manuscript (as shown in the Figure S4 below).*

[Figure]

Figure S3: Daily behavior of sea ice thickness and volume based on CS2SMOS dataset from October 2010 through April 2020. (a) Mean sea ice thickness within area of actual ice coverage. (b) Total(black),first-year(blue) and multiyear (red) sea ice volumes within Arctic basin. The mean sea ice thickness is computed within the area of actual ice coverage bounded by the gateways into the Pacific (Bering Strait), the Canadian Arctic Archipelago, and the Greenland (Fram Strait) and Barents Seas.

[Figure]

Figure S4: Interannual changes in sea ice volume, area and thickness based on the ICESat (2003–2008) and CryoSat-2 (2011–2020) satellite datasets. (a) Mean sea ice thickness within area of actual ice coverage. (b) Total sea ice area (cumulative area of actual ice coverage) within Arctic basin. (c) Total(black),first-year(blue) and multiyear (red) sea ice volumes within Arctic basin. Arctic basin volume and area is computed within the bounded by the gateways into the Pacific (Bering Strait), the Canadian Arctic Archipelago, and the Greenland (Fram Strait) and Barents Seas.

3) Finally I'm not really sure what the whole-Arctic mean SIT minimum is supposed to tell us. I can imagine a year where there's an early freezeup and therefore a lot of very thin FYI coverage. So volume could be up, but mean SIT down. (but what if this weren't measurable by CS2?) But I can also imagine a scenario where we're low on MYI, so volume could be down as well as mean SIT. So does whole-Arctic mean SIT mean anything? I think some more consideration of the relationship between the metric and sea ice volume is warranted. (for instance just before the freeze-up there is less sea ice volume than afterwards. But thin FYI proliferates after the freeze-up. So the effect of the freeze-up (I think) is that sea ice volume goes up, but mean sea ice thickness goes down sharply. But then

volume and mean-SIT both grow together). So the minimum occurs after freezeup but before the ice starts thickening in earnest. Does the day of minimum mean-SIT therefore occur before CS2 starts working? I'd be interested to know on what day of the year piomas predicts the minimum in whole-Arctic mean SIT, and for that matter what year has the lowest minimum in that data.

*Response: This point is well taken. As shown in figure S3 and S4, there is no 'minimum' visible in the SIT timeseries. This suggests that the minimum occurs before CS2 starts working. So what we have really identified is a record low for October, not an absolute record low. We have refined the title of our manuscript as: On the 2011 Arctic sea ice thickness: a combination of dynamic and thermodynamic anomalies.*

*This figure as shown in the Figure S5 below shows the Interannual changes in mean sea ice thickness based on the PIOMAS (2011–2016). PIOMAS shows that the daily minimum in whole-Arctic mean SIT at the end of October in 2012(0.908m), while the minimum mean SIT in 2011 is on November 1st with 0.911m. The day of PIOMAS predicts the minimum in whole-Arctic mean SIT is disagree with the CS2 and CS2SMOS.*

*The influence of sea ice is not just about how much ice volume there is. It is about how that ice is distributed. In 2011 the CS2 data suggest that the ice is unusually thinly distributed, and so that justifies a closer examination of what happened in that year. Sea ice thickness (SIT) is a key factor in the study of sea ice variability and their feedback effects, and it can better represent the mass balance of sea ice. Thicker sea ice is more insulated, weakening the coupling between the ocean and the atmosphere. It can limit the heat transfer from the ocean to the atmosphere in winter and affecting the thermodynamics of sea ice. SIT can also control the dynamics of ice; For example, SIT determines whether the ice is ridged or rafted. Finally, thick ice is more likely to survive the melting season, providing an opportunity to predict sea ice status on a seasonal time scale.*

*The reviewer is broadly commenting that a low SIT anomaly can arise in several ways. We agree with that entirely, which is why we chose to closely examine precisely why the SIT metric was low in this particular year.*

[Figure]

Figure S5: Interannual changes in mean sea ice thickness based on the PIOMAS (2010–2016).

A couple of narrower points:

1) The authors state: "The mean sea ice thickness within the area of actual ice coverage in October 2011 reached the lowest record for that calendar month in any year of the satellite records". The authors should point out that they have not examined the whole satellite record, which includes pan-Arctic SIT snapshots from ICESAT (2003-2010), and coverage up to 81.5 degrees by Envisat. Would be perhaps worth confirming that 2011 is a record low when also compared to ICESAT derived thickness? In particular I'm thinking about winter 2007-8 after the SIE minimum.

***Response***: *This is an important point raised by the reviewer and reminds us to check whether 2011 is a record low when compared to ICESAT derived thickness. We have now added a new figure to the Supplement of the revised version of our manuscript (as shown in the Figure S4 below) and compared the sea ice volume, area and mean sea ice thickness based on the ICESat (2003–2008) and CryoSat-2 (2011–2020) satellite datasets The fact that Kwok (2018) also has a paper published discussing Arctic sea ice volume in 2011  hit the lowest record from 0ct. to Nov. between 2003 and 2018 in the same Arctic basin means that i) the satellite data are considered worthy of studying and ii) this individual event is worthy of studying.*

[Figure]

Figure S4: Interannual changes in sea ice volume, area and thickness based on the ICESat (2003–2008) and CryoSat-2 (2011–2020) satellite datasets. (a) Mean sea ice thickness within area of actual ice coverage. (b) Total sea ice area (cumulative area of actual ice coverage) within Arctic basin. (c) Total(black),first-year(blue) and multiyear (red) sea ice volumes within Arctic basin. Arctic

basin volume and area is computed within the bounded by the gateways into the Pacific (Bering Strait), the Canadian Arctic Archipelago, and the Greenland (Fram Strait) and Barents Seas.

2) I'm also not sure that it's right to cite the NSIDC Kurtz & Harbeck data as an ESA product. Given I think both Kurtz and Harbeck were at and still do work at NASA? Could be wrong about this though.

*Response: Thanks for this comment. We refined this description as: To evaluate sea ice variability, we use two CryoSat-2 ice thickness products from 2010 to 2020 and ICESat ice thickness products from 2003 to 2008. The NASA GSFC (Goddard Space Flight Center) CryoSat-2 ice thickness data are available from NSIDC on a 25km polar stereographic grid (Kurtz and Harbeck, 2017). The AWI CryoSat-2 ice thickness data on a 25km EASE2 grid are available at Meereisportal (Ricker et al., 2014). The NASA GSFC CryoSat-2 ice thickness data provided daily from October 2010 to April 2020, while the AWI CryoSat-2 ice thickness provided weekly.*

3) Figure 4(b): It looks a lot like FYI export from the FS is negative for almost all months here? So it's flowing Northwards? Maybe I have the sign convention wrong, but in that case isn't MYI flowing backwards? I think some explanation is warranted about why it looks like there's an ice-type-dependent flow direction.

*Response: Thanks for this comment. Figure 4b has shown the monthly mean arctic sea ice export anomaly at Fram Strait, compared with the average from 2010 to 2016. The positive anomaly represents that more ice was exported at Fram Strait. As shown in Table 1, the FYI、MYI and total sea ice export from the Fram Strait were flowing southwards.  Compared with the average from 2010 to 2016, the amount of sea ice exported through the Fram Strait was occupied by more MYI, resulting in the corresponding negative anomaly of FYI.*

4) Figure 5): "Wind anomalies". Does the length of the arrow represent the magnitude of the velocity vector anomaly? Or the magnitude of the wind speed anomaly? These can be quite different. If it's the first then a large arrow can represent wind of the same speed going in a very different direction. If it's the second, then a large arrow can represent wind blowing in the same direction but at a different speed. I suppose it must be the first, because you can't have a negative arrow size? Or maybe it could be, because the arrows could then point backwards. Worth clarifying.

*Response*: *Thanks for this comment. The wind anomalies in Fig.5 is computed by the 10m wind velocity vector from monthly ERA5 atmospheric reanalysis data. We have carefully checked through this manuscript and rephrased the caption of Fig.5 in the revised: Winds velocity vector anomalies at 10m (m/s) and sea level pressure anomalies (hPa) (bottom row). Positive values indicate convergence, while negative values indicate divergence.*

5) Lastly since this work presents data from CS2 altimetry and a model (which assimilates a related product: the CS2-SMOS data), as a reader I'm interested to know how independent the model and the altimetry are. Does the SIT data 'force' the model behaviour? Or is it a weak influence that can be relatively ignored?

*Response*: *Thanks for this comment.*

*CS2 data have relatively large errors over the thin ice area, while SMOS has smaller error, and vice versa for the thick ice area. The complementary character between CS2 and SMOS motivates this product as pointed in Mu et al. (2018). CMST assimilates CS2 and SMOS sea ice thickness data. So we replaced the data with CS2SMOS for consistency to compare the daily behavior of sea ice thickness and volume from October 2010 through April 2020 and calculated the uncertainties in SIT as shown in the Figure S2.*

*For a data assimilation (DA) system, at the very beginning, the model generally has systematic errors and deviates from the observations. DA will push the model close to the observation during the so-called "spin-up" period, normally one or two weeks, which depends on the inflation of the ensemble. This process serves as an initialization of the DA system but in a modest way. In that sense, the SIT data force the model. After this "spin-up" period, the model is quite close to the observation. Most of the SIT variations can be simulated now. However, due to systematic error again, there are minor differences still between model simulations and observations. We reconcile the model errors (note that we have an ensemble) and observations errors to obtain the state with the maximum likelihood. During this period, both the model and DA play important roles, but it is difficult to distinguish their effects.*